# CLIQ: CONTRASTIVE LEARNING WITH XAI-GUIDED INTERPRETATION AND MODEL QUANTIZATION FOR EEG-BASED EMOTION RECOGNITION

## ABSTRACT

Electroencephalogram (EEG) may be a promising way to recognize human emotions in contrast to outward expressions, which may be hidden or artificially simulated. This paper applies self-supervised learning (SSL) to process complex EEG signals with low amount of labeled data for solving emotion recognition task. Proposed approach is based on a convolutional encoder with a novel contrastive loss and batching function. It has been evaluated on SEED and DEAP datasets. We also compared different preprocessing techniques in temporal, frequency and temporal-frequency domains. We achieved fairly high accuracy even on small amount of labeled data with the best accuracy of 88.7% and 87,3% on SEED, and 95.3% and 63.1% accuracy on DEAP for subject-dependent and subject-independent evaluations, respectively. Additionaly, we performed feature analysis and found that the greatest inter-emotional difference was shown in the T7 and T8 channels. We validated these findings with an iterative application of DeepLIFT. Combined with model quantization, these insights enabled us to reduce data and model size without significant decrease of accuracy. The proposed approach achieved separable vector representations of EEG and performance compatible with SOTA, enabled insightful data analysis, model interpretation with reasonable data reduction, and efficient model quantization.

## 1 INTRODUCTION

Emotions represent complex automatic physiological reactions triggered by internal cognition or external stimuli (Damasio, 1999; Ekman, 1992). They play a significant role in health, decision making, social interaction, and other aspects of human daily life (Bechara, 2005; Anderson et al., 2011). Therefore, emotion recognition (ER) may have applications in different fields both commercial and healthcare domains (Vecchiato et al., 2014; Anderson et al., 2011). Emotions may be recognized through various sources, including facial expressions and gestures, speech and voice, and physiological signals, such as electroencephalogram (EEG) and electrocardiogram (ECG) (Al-Nafjan et al., 2017). Although outward expressions of emotion may be hidden or artificially simulated, physiological signs, such as EEG, will remain stable, making predictions based on them more reliable and robust.

EEG is a non-invasive neurophysiological functional imaging technique that is recorded from the surface of the head to measure electrical brain activity (Teplan et al., 2002). Due to the fact that EEG signals contain information about the brain's response to various internal and external stimuli, as well as its non-invasive nature, accessibility and high temporal resolution (Jiang et al., 2019), EEG is widely used in different real-world domains, including emotion recognition (Bos et al., 2006). During recordings and experiments, EEG signals have low amplitude, therefore they are affected by noise and various artifacts, such as participants' blinks or movements, and potential scalp-electrode contact imperfections (Sheoran et al., 2015). This, as well as the complex structure of the EEG signals, causes classical statistical and regression approaches to struggle with extraction of complex nonlinear patterns in EEG signals (López-García et al., 2020; Tzimourta et al., 2021). Therefore, researchers have applied deep learning techniques to process EEG signals (Acharya et al., 2018; Craik et al., 2019).

Deep learning models showed significant results in EEG signals classification (Siuly et al., 2016; Craik et al., 2019), regression (Sabbagh et al., 2020), and generation (Fahimi et al., 2019) tasks, in particular for the recognition of emotions (Palo et al., 2015). Such architectures as Long-Short Term Memory Network (LSTM), Graph Neural Network (GNN), Transformer and Convolutional Neural Network (CNN) (Vaswani, 2017; Joshi et al., 2022) allow to capture complex spatial-temporal information underlying brain dynamics. However, a number of potential problems arise from the processing of EEG signals by deep supervised learning models: **Expert-Intensive EEG Labeling**: deep models require significant amounts of labeled data for good generalization and effective training. Therefore, experts with professional knowledge and extensive experience in neurophysiology have to label data that require a significant amount of staff time (Liu & Fu, 2021). **EEG signals complexity:** processing of EEG signals, presented as long multi-channel sequences with complex nature, result in generalization failure (Adeli & Wu, 1998), unreliable estimations or convergence in significant time (Hinton & Salakhutdinov, 2006) for supervised learning models and for some tasks only marginally outperform random predictions (Xiao et al., 2024). One promising approach that has already shown high potential for natural language and image processing and is capable for extracting representations from unlabeled data is self-supervised learning (SSL) (Weng et al., 2024), which show high performance even on small portion of labeled data (Rafiei et al., 2022; Banville et al., 2021). In particular, applying contrastive learning makes similar data samples closer to each other and separates different samples in the embedding space.

A perspective development in the field of emotion recognition is real-time processing and prediction on lightweight devices. In particular, the model may be able to adapt to a specific user through online learning. However, the limited computational resources of lightweight devices poses a demand for compact and efficient models that are capable of both maintaining high prediction accuracy and occupying a minimum of device memory. For this purpose, it is essential to perform features pruning and model compression techniques, such as quantization, which allow to optimize the approach and reduce the model size without significant loss of prediction accuracy (Khan et al., 2024). However, frequently current EEG-based emotion recognition methods employ complex approaches suffer from several drawbacks including **limited interpretability, large model size and computational inefficiency** (Liao et al., 2024; Kan et al., 2023; Wan et al., 2023).

To address all described issues, we introduce CLIQ, a contrastive learning approach with novel pairing and batching techniques for emotion recognition by EEG. Additionally, we provide feature analysis and iterative DeepLIFT (Shrikumar et al., 2017) interpretation to identify T7 and T8 EEG channels with highest inter-emotional difference, then we apply model quantization, reducing both data and model size without significant accuracy decrease. Our detailed contributions are as follows:

- We introduce a **convolutional EEG model**, called **CLIQ**, for EEG decoding for emotion recognition application. For extraction of effective dependencies inside EEG signals we utilize various preprocessing techniques in three perspectives: temporal, frequency, and temporal-frequency domains. CLIQ is pre-trained on **SEED** and **DEAP** datasets for learning generic representations through contrastive learning with **a novel pairing, hard-soft negative contrastive loss and batching function**.

- We propose to handle negative pairs differently by splitting them on **soft and hard negative pairs** thus giving the model idea of more complex correspondence between stimulus. Moreover, we propose a new **batching technique** that strict the samples to be valuable, because batch from samples forming only positive pairs incorporates bias with respect to negative pairs in the loss and vise versa.

- We performed features analysis in order to identify EEG channels with greatest inter-emotional difference. Additionally, we used **DeepLIFT** to provide interpretation of CLIQ's predictions and prove channels identified by features analysis. As a result, we reduced the number of the EEG channels without significant loss of accuracy. We also perform an **inter-dataset transfer learning** on identified channels pre-training on SEED and fine tuning on DEAP and vise versa.

- We applied **symmetric post-training quantization** on presented model lowering the model size without significant loss of accuracy. The transition from storing weights and activations in float-point numbers to integers also allows to speed up the inference time.

## 2 METHOD

In this work, we propose CLIQ, an approach to classify emotions based on EEG using convolutional encoder (Figure 1). It is trained by contrastive learning on the unlabeled data during downstream task and then fine-tuned on the real labels. After training we perform feature analysis, interpret the results by XAI methods and quantize the final model.

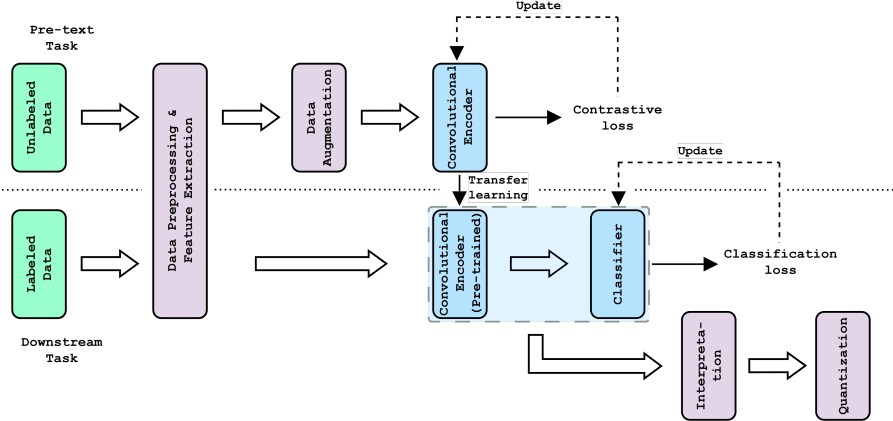

Figure 1: General pipeline of the proposed solution.

### 2.1 FEATURE EXTRACTION

**Baseline Correction.** A per-second average of resting-state signal was subtracted from the stimulated recording, which was then standardized along the time axis. **Spatial Transformation.** The original signal was mapped to $9X9$ matrix (Figure 6) with channels located according to 10-20 system at each moment of time, making the data to have one more dimension. **Power Spectral Density (PSD).** PSD of a raw signal was calculaculated by Welch's method. **Differential Entropy (DE).** DE features of a spectral representation of a signal, obtained by FFT with a Hanning window, were calculated in 5 frequency bands: delta (1-3 Hz), theta (4-7 Hz), alpha (8-12 Hz), beta (13-30 Hz), gamma (31-50 Hz). **DASM, RASM.** These features were defined as difference and ratio of the DE features for 14 pairs of asymmetric electrodes in left and right hemispheres. **DCAU.** DCAU feature is defined as the difference of the DE features for 11 pairs of frontal-posterior electrodes. More details with mathematical formulation described in A.1. **Data augmentations.** During this study we used 3 wide-spread data augmentation techniques, which are jittering, samples mix-up, and channel masking.

### 2.2 CONTRASTIVE FRAMEWORK

**Pairs formation.** In this research we propose an idea of soft and hard negative pair that splits not so different pairs from the negative ones. As a result, we set recordings of one subject having the same stimulus with its augmentations as positive pairs, different subjects having the same stimulus and their augmentations as soft negative pairs, and other combinations as hard negative pairs.

**Batching.** We follow the idea of lifted structure loss, where pairs are formed as combinations of items in the batch. We propose an algorithm for batching formation, which ensures that each element in the batch has at least two different items forming with it a positive and a negative pair respectively.

Let the group be a list of samples that could form a positive pair with any other element of this group. First, shuffle all data elements. Then select any group and pick a random number $m$ from the uniform distribution between 2 and $max(3, min(b, r)//k)$, where $b$ is a batch size, $r$ is a number of free spots in a batch, and $k$ is a parameter to handle variability of the dataset, set to 2. Randomly select $m$ items from the group and put in the batch. One may check the number of remaining items in the group and if it is less than 2, put these elements in the batch. After that select another group and repeat iterations until all samples will be distributed among batches. Note that the batch size

should be larger than 3, and some batches may vary in size unless stricter conditions have been established.

**Contrastive loss.** We adopt a normalized temperature-scaled binary cross-entropy loss to handle soft and hard negative pairs in a different proportion.

$$l_{ij} = -\alpha y \log \sigma(f_{ij}) - \beta(1-y)\delta \log(1 - \sigma(f_{ij})) - \gamma(1-y)(1-\delta)\log(1-\sigma(f_{ij})) \quad (1)$$

where $f_{ij}$ is a temperature scaled cosine similarity between embeddings of samples $i$ and $j$, $\sigma$ is a sigmoid function, $\alpha$, $\beta$, $\gamma$ are parameters between 0 and 1, which regulate the contribution of positive, soft and hard negative pairs, $y$ could take values of 0 or 1 depending on whether $i$ and $j$ are a negative or positive pair, and $\delta$ represents whether a pair is soft or hard negative.

**Model architecture.** Encoder consists of several 2-dimensional convolutional layers, followed by activation function, pooling, batch normalization, and dropout. Then flattening and linear mapping were applied. The projector follows encoder during downstream task and consists of linear layers separated by Leaky ReLU function, batch normalizetion and dropout. More detailed architecture described in A.2.

## 2.3 INTERPRETATION

**Features analysis.** For the analysis we used DE features, which were grouped into distinct emotional states (positive, negative, neutral), and for each channel were averaged by participants, videos, trails, time, and frequency bands to form a single value for each emotional state and channel. The resulting values were mapped to the channel name according to the 10-20 system on the electrode location map and normalized across all values. In addition, the absolute difference of the obtained maps between pairs of emotional states (positive-negative, positive-neutral, negative-neutral) was calculated and then normalized.

**Iterative Application of XAI method.** To interpret predictions made by the EEG-based emotion classification model we decided to implement Deep Learning Important FeaTures (DeepLIFT) XAI technique overviewed in (Shrikumar et al., 2017).

We propose an iterative application of DeepLIFT methodology across different experimental scenarios for an interpretation analysis. The primary goal of these iterative applications is to systematically identify and remove EEG channels and frequency bands with minimal or irrelevant contributions. By removing unnecessary EEG features, we significantly improve the efficiency of model training and inference, reducing computational load and model complexity without compromising predictive performance.

## 2.4 QUANTIZATION

Since the model would be trained using SSL, we applied post training quantization (PTQ) technique to reduce it. As symmetric absolute maximum linear mapping is less computationally expensive and provides better precision we used it according to the following formula:

$$s = \frac{2^{b-1}-1}{\alpha} \quad (2)$$

$$x_{quantized} = round(s \cdot x) \quad (3)$$

$$x_{dequantized} = \frac{1}{s} \cdot x_{quantized}, \quad (4)$$

where $s$ is the calculated scaling factor, $b$ is the number of bits in integer quantized range, $\alpha$ is the absolute maximum value of the quantized data, $x$ is value to be quantized, $x_{quantized}$ and $x_{dequantized}$ are quantized and dequantized values respectively.

Since the model weights are static in the inference stage, their quantization parameters were computed single time. Activations vary with the input data, so static quantization with Min-Max calibration was chosen for them. During calibration, the maximum value of the activation tensor for each model layer was collected from the calibration dataset to calculate the scale by equation 2. Meanwhile, one used per-channel granularity for the weights, and per-tensor granularity for the activations.

# 3 EXPERIMENTS

## 3.1 DATASETS

For recognizing human emotional states, many various datasets exist. Two commonly applied datasets, SEED and DEAP, are used in this paper.

**SEED** was collected by the Brain-like Computing and Machine Intelligence (BCMI) laboratory (Zheng & Lu, 2015). The experimental stimuli were 15 videos with 3 types of emotions - positive, neutral and negative - forming the corresponding labels. EEG signals were collected using a 62-channel ESI NeuroScan System with a sampling rate of 1000 Hz. When processing the full video, the data contains EEG recordings of 15 people who watched 15 movies of 3-4 minutes length 3 times each, thus the dataset contains 675 data samples. When the data were divided into 1 second segments, the size of the dataset increased accordingly. Raw EEG data were downsampled to a sampling rate of 200 Hz. One applied a band-pass frequency filter from 0 to 75 Hz.

**DEAP** is a multimodal dataset for the analysis of human affective states (Koelstra et al., 2011). EEG and peripheral physiologic signals of 32 participants were recorded while watching 40 one-minute segments of videos. Participants rated each video in terms of the levels of arousal, valence, dominance, like/dislike and familiarity. During this study, we formed a positive and negative labels of emotions according to the valence label and selected only EEG signals from the dataset. The raw DEAP data was downsampled to a sampling rate of 128 Hz. One applied a band-pass frequency filter from 4 to 45 Hz, and removed EOG artefacts (Koelstra et al., 2011).

## 3.2 EXPERIMENTAL SETUP

**Hyperparameters of contrastive loss** explained by Equation 1 were set as follows: $\alpha = \beta = 0.4$, because we consider positive and hard negative pairs equally important, and $\gamma = 0.2$ to penalize soft-negative pairs twice less than hard-negative pairs. The temperature scale $\tau$ was established to 0.5.

**Hyperparameters for data augmentation**s were selected according the recommendation in the existing approaches (Liao et al., 2024). The jittering values were sampled from the Gaussian distribution with mean 0 and variance 0.2. The number of channels for masking is chosen from a uniform distribution between 1 and a maximal number of channels that could be zeroed (at least 2). The proportion for samples mix-up was chosen from the uniform distribution between 0 and 0.5.

**The early stopping technique** was incorporated to prevent the overfitting. The model weights are saved, and the counter is reset to zero as soon as the test loss calculated on the current epoch is less than 99.9% of the best previous iteration loss. Otherwise, the counter is incremented by 1. If the counter exceeds the threshold value, the process is stopped.

During **the pre-text task**, training data was batched using the technique described in Section 2.2. Testing set was split into batches once without shuffling. The counter threshold value set to 25 epochs. Generally, the training on pre-text tasks runs for 300 epochs, however, basically the early stopping criteria work after 50-150 epochs decreasing the resource consumption.

During **the downstream task**, training data were batched using the default PyTorch batching function with random shuffling, while the testing data were the same as in the pre-text task, i.e. never seen before by the model. The loss used for emotion classification is a Cross Entropy Loss with default setup. The early stop threshold was set to 15 epochs. Generally, the training on downstream tasks runs for 200 epochs, however, usually the early stopping criteria is triggered after 25-60 epochs decreasing the resource consumption.

All experiments were run on Tesla T4 GPU. Also, every batch size was set to 128 and we used Adam optimizer with cyclic learning rate (LR) scheduler with base LR set to 0.001, maximal LR - 0.1, and exponential range mode. The code was written in python programming language with the usage of PyTorch, Pandas, NumPy, MatPlotLib, Scikit-learn and Seaborn libraries.

**Evaluation.** We applied two approaches to evaluate the ability of the model to predict emotions based on EEG data. Subject-dependent (SD) split assumes that data of each subject is presented both in training and testing sets, while subject-independent (SI) evaluation insists having in the test

set data of unseen subjects. For comparison with existing approaches we extend the evaluations to 10-fold cross validation and Leave-One-Subject-Out Cross-Validation accordingly.

**Baselines.** We compare CLIQ with on emotion recognition task on SEED and DEAP datasets for both subject-dependent and subject-independent evaluations. We consider the following main methods: **TS-MoCo** (Hallgarten et al., 2023), **CLDTA** (Liao et al., 2024), **SGMC** (Kan et al., 2023), **GMSS** (Li et al., 2022), **EEGformer** (Wan et al., 2023), **MMResLSTM** (Ma et al., 2019), **RGNN** (Zhong et al., 2020), **MSBAM** (Wu et al., 2022). The model size for each existing approach was evaluated based on their description from source papers or the provided code implementation.

### 3.3 EXPERIMENTAL RESULTS

**Comparison with baseline.** To show that the proposed solution achieved good results, we compared the accuracy (in %) of our approach with the state-of-the-art (SOTA) models, both in FSL and SSL modes. The comparison of the best solutions found for SEED is shown in Table 1 and for DEAP in Table 2. The tables present the accuracy provided by the authors in the source articles. If the corresponding metric was not found in the original article, the accuracy reported by other authors applying this method was used. If this metric was still missing, a dash was written. For the comparison, our model was taken with the parameters described above and below, and specifically for the SEED dataset - DE features, for the DEAP dataset - baseline correction.

Table 1: Comparison of CLIQ with existing solutions on SEED

| Model | Mode | SI | SD | Size | Memory |
|---|---|---|---|---|---|
| SVM (Li et al., 2022) | FSL | $56.73 \pm 16.29$ | $83.99 \pm 9.72$ | – | – |
| A-LSTM (Song et al., 2019) | FSL | $72.18 \pm 10.85$ | $88.61 \pm 10.16$ | – | – |
| DGCNN (Song et al., 2018) | FSL | $79.95 \pm 9.02$ | $90.40 \pm 8.49$ | – | – |
| EEGFormer (Wan et al., 2023) | FSL | – | $91.58 \pm 2.77$ | 20 M | 80 MB |
| RGNN (Zhong et al., 2020) | FSL | $85.30 \pm 6.72$ | $94.24 \pm 5.95$ | 2.5 M | 10 MB |
| GMSS (Li et al., 2022) | FSL | $\mathbf{86.52 \pm 6.22}$ | $\mathbf{96.48 \pm 4.63}$ | 12.5 M | 50 MB |
| TS-MoCo (Hallgarten et al., 2023) | SSL | – | 43.00 | 8.5 M | 34 MB |
| SSL-EEG (Li et al., 2022) | SSL | $67.52 \pm 12.73$ | $83.32 \pm 9.20$ | – | – |
| CLDTA (Liao et al., 2024) | SSL | $75.09 \pm 5.88$ | $93.12 \pm 5.02$ | 11 M | 44 MB |
| SGMC (Kan et al., 2023) | SSL | – | $\mathbf{94.04}$ | 27.5 M | 110 MB |
| GMSS (Li et al., 2022) | SSL | $76.04 \pm 11.91$ | $89.18 \pm 9.74$ | 12.5 M | 50 MB |
| SeqCLR (Mohsenvand et al., 2020) | SSL | 78.40 | 85.77 | – | – |
| CLISA (Shen et al., 2022) | SSL | $86.40 \pm 6.30$ | – | – | – |
| CLIQ (ours) | SSL | $\mathbf{87.28 \pm 4.5}$ | $88.70 \pm 2.98$ | 3 M | 10 MB |
| CLIQ (T7,T8, quantized) | SSL | $\mathbf{85.3 \pm 4.5}$ | $83.8 \pm 2.98$ | 2.5 M | 2.5 MB |

According to the obtained results, the proposed approach achieved comparatively high performance relative to SOTA models. More precisely, the proposed approach outperforms existing solutions in subject-independent evaluation for SEED and subject-dependent evaluation for DEAP.

#### 3.3.1 ABLATION ON HARD-SOFT NEGATIVE PAIRING

During this study we evaluated the effect of proposed hard and soft negative pairs, the corresponding modification in the contrastive loss function and the incorporated batching technique. We compared performance of our solution with the basic one, meaning only positive and negative samples, basic normalized temperature-scaled cross entropy loss with batching by random shuffling, similarly to the solutions presented by (Liao et al., 2024).

From the comparison in Figure 3, one can conclude that the proposed hard-soft negative pairing with modified loss and special batching have positive impact on the accuracy across both datasets with DE and baseline feature extraction techniques.

Table 2: Comparison of CLIQ with existing solutions on DEAP

| Model | Mode | SI | SD |
|---|---|---|---|
| RODAN (Lew et al., 2020) | FSL | $56.8 \pm 3.3$ | $85.4 \pm 0.3$ |
| 3DCNN (Salama et al., 2018) | FSL | 60.7 | 87.4 |
| LSTM + RAW (Chen et al., 2019) | FSL | 63.7 | – |
| H-AVE-BGRU + RAW (Chen et al., 2019) | FSL | 65.8 | – |
| DCCA (Lan et al., 2020) | FSL | – | $85.6 \pm 3.48$ |
| ECLGCNN (Yin et al., 2021) | FSL | – | $90.5 \pm 3.09$ |
| MMResLSTM (Ma et al., 2019) | FSL | – | $92.3 \pm 1.55$ |
| BiDCNN (Huang et al., 2021) | FSL | **68.1** | **94.4** |
| Cascaded SSL (Wang et al., 2024) | SSL | **$65.5 \pm 5.47$** | – |
| GANSER (Zhang et al., 2022) | SSL | – | 93.5 |
| CLDTA (Liao et al., 2024) | SSL | – | $94.6 \pm 1.40$ |
| SGMC (Kan et al., 2023) | SSL | – | 94.7 |
| CLIQ (ours) | SSL | $63.1 \pm 9.85$ | **$95.3 \pm 4.80$** |

Table 3: Influence of soft-hard negative pairs, modified loss and batching on the performance.

| Approach type | SEED DE | | DEAP Baseline | |
|---|---|---|---|---|
| | SI | SD | SI | SD |
| Basic | 86.4 | 82.3 | 57.5 | 94.5 |
| Proposed | 87.3 | 88.7 | 63.1 | 95.3 |

We want to note that the soft and hard negative pairs could provide more insights about complex connections of the stimulus.

On another hand, without an appropriate batching technique, batch could contain non-representative combinations (e.g. only positive), which incorporates biases and lowers robustness.

### 3.3.2 VALIDATION OF THE PRE-TRAINING RESULTS

Since SSL helps to train the model on a small amount of labeled data, we analyzed the dependence of accuracy on the amount of labeled data in the training set (Figure 2). We conducted experiments with training the encoder on a pre-text task and freezing its weights with DE on SEED dataset and Baseline correction for DEAP dataset. The downstream task was performed on 8%, 25%, 50%, 75% and 100% of the SEED training data with subject-independent evaluation, and on on 1%, 10%, 25%, 50%, 75% and 100% of the DEAP training data with subject-dependent evaluation. For comparison, exactly the same model was trained on the same data in a fully supervised mode without layers freezing. The proposed approach showed high stable accuracies on the small amount of training data both for SEED and DEAP, supporting the efficiency of the obtained embedding.

To visually assess the ability of the pre-text task to separate embeddings, we applied t-SNE and PCA to compress the embeddings obtained for the samples from test set into 2 dimensions, which are shown in Figure 2. Same colors represent elements that form positive pairs. As a results of visual assessment one can conclude that points of the same color became closer to each other and father apart other groups.

### 3.3.3 RESULTS OF INTERPRETATION

**Feature analysis and pruning.**

From the constructed maps, one found that the greatest inter-emotional difference was shown in the T7 and T8 channels.

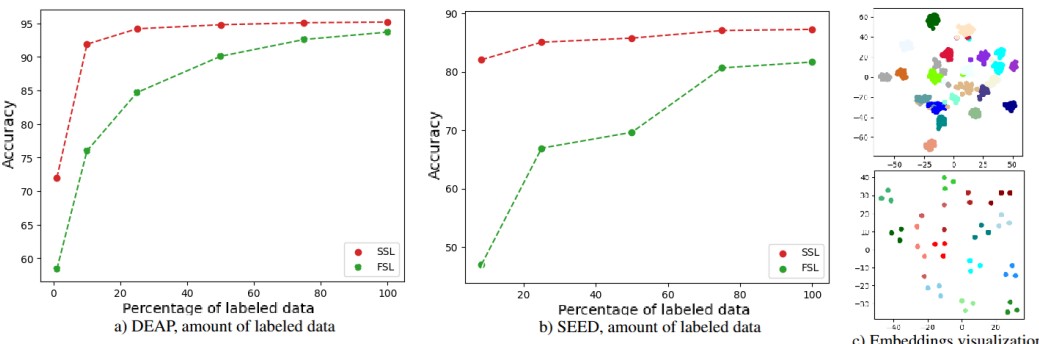

Figure 2: Left - Accuracy dependence of SSL and FSL modes on the amount of labeled data. Right - Visualization of embeddings compressed to 2 dimensions using t-SNE on DEAP (up) and SEED (down).

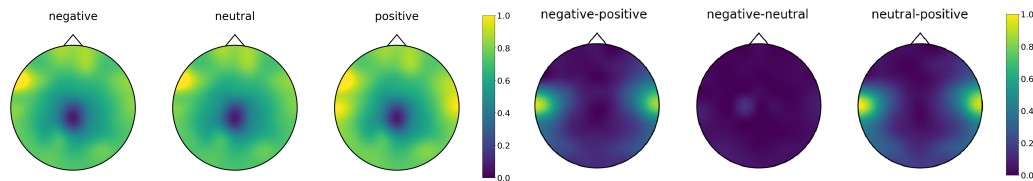

Figure 3: Averaged DE features of SEED mapped by electrodes places according to 10-20 system with difference between distinct pairs of emotional states.

Nevertheless, we decided to perform feature pruning of the initial data for SEED and DEAP datasets on DE, keeping only channels T7 and T8. Due to the reduced dimensionality of the initial data, we modified the model architecture by reducing the kernel size inside the convolutional operations from $(3 \times 3)$ to $(1 \times 3)$. The modified architecture was trained with the same approach on the original number of channels as well as on channels T7 and T8. The results of the experiments performed and comparisons with the previous results are shown in Table 4.

Moreover, we performed inter-dataset experiments shown in Table 4, with the sample size cut to the smallest possible size. Specifically, for SEED2DEAP, we pre-trained the encoder on SEED, and then fine-tuned and adjusted the entire model on DEAP. For DEAP2SEED, we pre-trained the encoder on DEAP, and then fine-tuned and evaluated the entire model on SEED. In both cases for each dataset we used all availible data on each emotional state.

Table 4: Comparison of the accuracy of the proposed model to one with a smaller kernel size, on channels T7 and T8 and for transfer learning

| Data | Kernel shape | Channels number | DE | | Baseline | |
|---|---|---|---|---|---|---|
| | | | SI | SD | SI | SD |
| SEED | $(3 \times 3)$ | 62 | 87.3 | 88.7 | 63.9 | 91.2 |
| SEED | $(1 \times 3)$ | 62 | 86.6 | 87.2 | 53.9 | 90.9 |
| SEED | $(1 \times 3)$ | T7, T8 | 85.8 | 84.2 | 47.9 | 88.3 |
| DEAP2SEED | $(1 \times 3)$ | T7, T8 | 63.7 | 60.1 | 48.5 | 60.1 |
| DEAP | $(3 \times 3)$ | 32 | 60.6 | 62.4 | 63.1 | 95.3 |
| DEAP | $(1 \times 3)$ | 32 | 60.6 | 61.8 | 51.6 | 94.9 |
| DEAP | $(1 \times 3)$ | T7, T8 | 54.9 | 58.5 | 53.4 | 89.5 |
| SEED2DEAP | $(1 \times 3)$ | T7, T8 | 63.4 | 59.4 | 54.2 | 87.2 |

**DeepLIFT Iterative Data Reduction.** Applying the iterative data reduction with DeepLIFT, we measured accuracy and speed of model training in terms of the seconds per iteration (epoch) for each new set of EEG channels. The results for both SEED and DEAP datasets are shown in Figures

4 and 5. Therefore, we managed to significantly optimize the model training process, degrading accuracy slightly inst of the experiments.

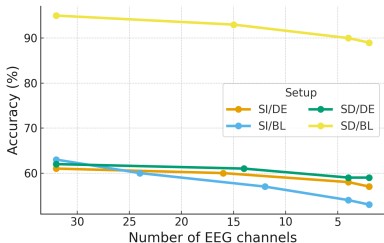 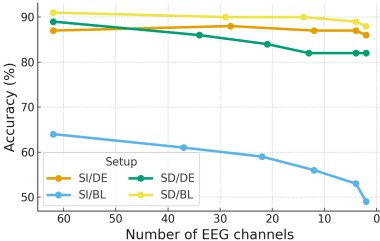

Figure 4: Accuracy (%) vs. Number of EEG channels for DEAP and SEED.

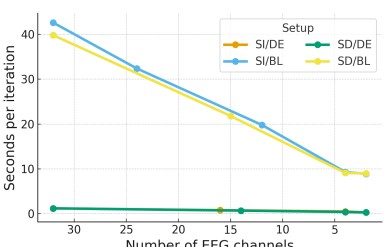 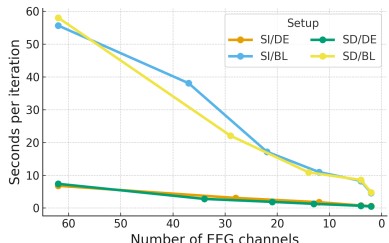

Figure 5: Model training speed on the pre-text task for DEAP and SEED: Seconds per iteration vs. Number of EEG channels.

### 3.3.4 RESULTS OF QUANTIZATION

After performing quantization, we evaluated the quantized model on T7 and T8 channels for SEED dataset DE features and obtained 85.3% for subject-independent and 83.8% for subject-dependent splits, what is comparable to the accuracy of the model before the quantization. Moreover, we compared the number of parameters and memory sizes occupied by our model with similar solutions for EEG-based emotion recognition using the SEED dataset. Result of comparison shown in Table 1. The comparison criteria for each existing solution architecture were evaluated based on their description from source papers or the official implementation.

The obtained quantization results allowed us to significantly reduce (by 4 times) the amount of memory occupied by the model computed for the SEED dataset. In addition, the transition from storing weights and activations in float-point numbers to integers allows to speed up the inference time from 13 sec to 3 sec for 135 samples from the test set.

## 4 CONCLUSION

In this paper, we propose a convolutional contrastive learning-based approach CLIQ for emotion recognition task that does not require a large amount of computational resources and utilize a new contrastive loss, pairing, batching function and various data preprocessing methods. We evaluate the proposed approach on SEED and DEAP datasets. The reached accuracy of predictions is comparable with existing approaches for both subject-dependent and subject-independent evaluations. Additionally, we performed feature analysis, DeepLIFT XAI interpretation and quantization technique, and reduced the number of the EEG channels and model size without significant loss of accuracy.

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

# A  APPENDIX

## A.1  FEATURE EXTRACTION TECHNIQUES

### A.1.1  SPATIAL TRANSFORMATION

The 3D representation of the original signal was formed according to $9X9$ matrix with channels located according to 10-20 system depicted in Figure 6 for 62 and 32 channels. To obtain the 3D representation of the whole signal we performed such a transformation at each time moment of the initial signal.

| 0 | 0 | 0 | FP1 | FPZ | FP2 | 0 | 0 | 0 |
|---|---|---|---|---|---|---|---|---|
| 0 | 0 | 0 | AF3 | 0 | AF4 | 0 | 0 | 0 |
| F7 | F5 | F3 | F1 | FZ | F2 | F4 | F6 | F8 |
| FT7 | FC5 | FC3 | FC1 | FCZ | FC2 | FC4 | FC6 | FT8 |
| T7 | C5 | C3 | C1 | CZ | C2 | C4 | C6 | T8 |
| TP7 | CP5 | CP3 | CP1 | CPZ | CP2 | CP4 | CP6 | TP8 |
| P7 | P5 | P3 | P1 | PZ | P2 | P4 | P6 | P8 |
| 0 | PO7 | PO5 | PO3 | POZ | PO4 | PO6 | PO8 | 0 |
| 0 | 0 | CB1 | O1 | OZ | O2 | CB2 | 0 | 0 |

| 0 | 0 | 0 | FP1 | 0 | FP2 | 0 | 0 | 0 |
|---|---|---|---|---|---|---|---|---|
| 0 | 0 | 0 | AF3 | 0 | AF4 | 0 | 0 | 0 |
| F7 | 0 | F3 | 0 | FZ | 0 | F4 | 0 | F8 |
| 0 | FC5 | 0 | FC1 | 0 | FC2 | 0 | FC6 | 0 |
| T7 | 0 | C3 | 0 | CZ | 0 | C4 | 0 | T8 |
| 0 | CP5 | 0 | CP1 | 0 | CP2 | 0 | CP6 | 0 |
| P7 | 0 | P3 | 0 | PZ | 0 | P4 | 0 | P8 |
| 0 | 0 | 0 | PO3 | 0 | PO4 | 0 | 0 | 0 |
| 0 | 0 | 0 | O1 | OZ | O2 | 0 | 0 | 0 |

Figure 6: Spatial transformation maps for 62 and 32 channel devices.

### A.1.2  DE

DE were calculated as follows:

$$DE(x) = 0.5 \log(2\pi e \sigma^2) \tag{5}$$

where $x$ is a time series with variance $\sigma^2$.

### A.1.3  DCAU, DASM, RASM

DASM and RASM features were defined as the difference and ratio of the DE features for 14 pairs of asymmetric electrodes shown in Table 5.

Table 5: Pairs of asymmetric electrodes for left (L) and right (R) hemisphere

| **L** | FP1 | F7 | F3 | T7 | P7 | C3 | P3 | O1 | AF3 | FC5 | FC1 | CP5 | CP1 | PO3 |
|---|---|---|---|---|---|---|---|---|---|---|---|---|---|---|
| **R** | FP2 | F8 | F4 | T8 | P8 | C4 | P4 | O2 | AF4 | FC6 | FC2 | CP6 | CP2 | PO4 |

DASM and RASM are calculated according to following formula:

$$DASM(x) = DE(x_{left}) - DE(x_{right}) \tag{6}$$

$$RASM(x) = \frac{DE(x_{left})}{DE(x_{right})}, \tag{7}$$

where $x$ is a signal, $DE(x_{left})$ and $DE(x_{right})$ are $DE$ of $x$ for left and right asymmetric electrodes respectively.

DCAU feature is defined as the difference of the DE features for 11 pairs of frontal-posterior electrodes shown in Table 6.

Table 6: Pairs of electrodes for frontal (F) and posterior (P) brain parts

| **F** | FC5 | FC1 | FC2 | FC6 | F7 | F3 | FZ | F4 | F8 | FP1 | FP2 |
|-------|-----|-----|-----|-----|----|----|----|----|----|-----|-----|
| **P** | CP5 | CP1 | CP2 | CP6 | P7 | P3 | PZ | P4 | P8 | O1 | O2 |

DCAU is calculated according to following formula:

$$DCAU(x) = DE(x_{frontal}) - DE(x_{posterior}), \tag{8}$$

where $x$ is a signal, $DE(x_{frontal})$ and $DE(x_{posterior})$ are $DE$ of $x$ for frontal and posterior pairs of electrodes respectively.

### A.2 MODEL ARCHITECTURE

#### A.2.1 ENCODER

Encoder architecture with its parameters is shown in the Table 7, where $k\_s$ is the kernel size, $in\_ch$ is the number of input channels, $out\_ch$ is the number of output channels, $pad$ is the type of padding, $p$ is the probability, $in$ is the size of the input vector, $out$ is the size of the output vector, $emb\_dim$ is the size of the embedding set to 128, $m$ is equal to 5 for time-frequency type of preprocessing and to 1 otherwise, $r$ is the length of the flattened data. In experiments, we also used the encoder with the $3 \times 3$ kernel in convolutions.

Table 7: Layout of encoder architecture with smaller kernel

| Layer | Parameters |
|-------|------------|
| Conv2d | $in\_ch = m, out\_ch = 64, k\_s = (1 \times 3), pad = (0,1,0,1)$ |
| LeakyReLU | |
| MaxPool2d | $k\_s = (1 \times 2), stride = (0,2)$ |
| BatchNorm2d | $in\_ch = 64$ |
| Dropout | $p = 0.25$ |
| Conv2d | $in\_ch = 64, out\_ch = 128, k\_s = (1 \times 3), pad = (0,1,0,1)$ |
| LeakyReLU | |
| MaxPool2d | $k\_s = (1 \times 2), stride = (0,2)$ |
| BatchNorm2d | $in\_ch = 128$ |
| Dropout | $p = 0.2$ |
| Flatten | |
| Linear | $in = r, out = emb\_dim$ |

#### A.2.2 PROJECTOR

The classifier sends the embedding obtained by the base model through a projector model, whose architecture with its parameters is described in Table 8, where $in$ is the size of the input vector, $out$ is the size of the output vector, $p$ is the probability, $emb\_dim$ is the size of the embedding (128).

Table 8: Layout of projector architecture with hyperparameters

| Layer | Parameters |
|-------|------------|
| Linear | $in = emb\_dim, out = 1024$ |
| LeakyReLU | |
| BatchNorm1d | $in = 1024$ |
| Dropout | $p = 0.3$ |
| Linear | $in = 1024, out = c$ |

## A.3 MORE DETAILS ON FEATURE ANALYSIS

We visualized the spectrogram of T7 and T8 channels for two arbitrary participants on arbitrary trails generated by sequentially concatenating all 15 viewed videos. In the obtained spectrograms, one may notice a clear difference during the transition between positive-negative and positive-neutral (Figure 7).

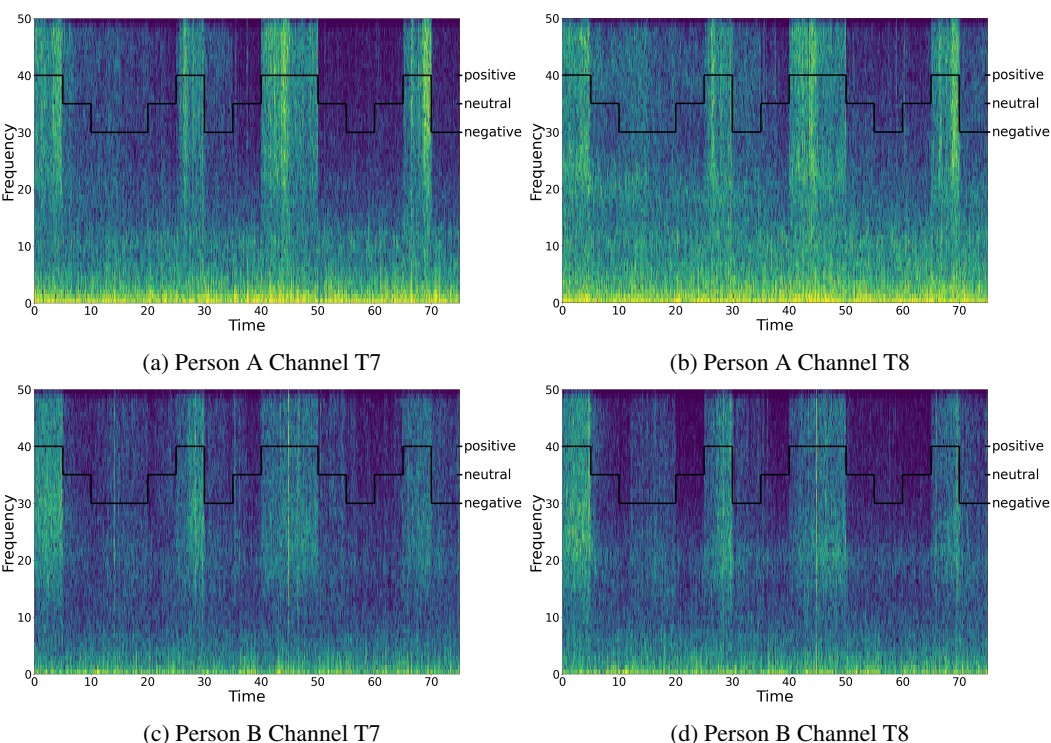

(a) Person A Channel T7

(b) Person A Channel T8

(c) Person B Channel T7

(d) Person B Channel T8

Figure 7: Visualization of SEED spectrograms highlighting variation of emotional states for different stimuli on T7 & T8 channels for 2 arbitrary subjects.

Following the obtained results, we decided to visualize similar averaged maps for the DEAP dataset that is shown in Figure 8.

## A.4 DEEPLIFT EXPLANATION

DeepLIFT is an advanced explainability method designed to measure feature contributions by comparing the model's output between a given input and a carefully chosen reference baseline (commonly zero-level input). DeepLIFT calculates feature attribution scores based on the differences in activation across neurons when processing the input data versus the baseline (Shrikumar et al., 2017).

Formally, DeepLIFT defines contribution scores through differences from a baseline input. Let be the actual input and be the baseline input. If the model output is denoted as , the contribution of the input feature is computed as:

$$C_{\Delta x_i} = (x_i - x_i^0) \frac{\partial f}{\partial x_i}\bigg|_{\text{baseline}} \qquad (9)$$

For convolutional layers, DeepLIFT calculates attribution by propagating the activation differences backward through convolutional filters. Consider a convolutional neuron output computed from inputs (spatial region) with convolutional weights . The neuron's output difference relative to the baseline is:

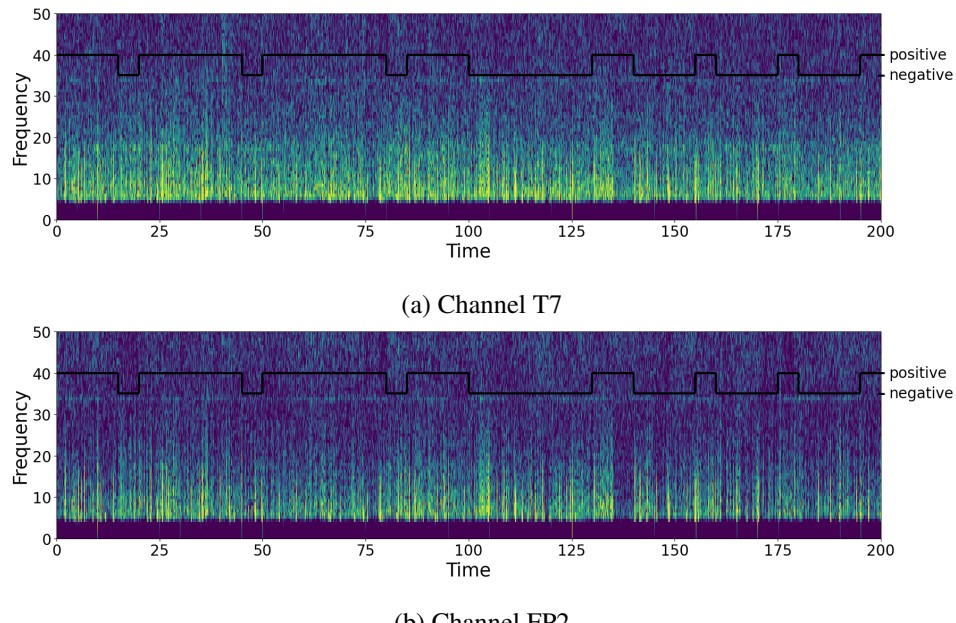

(a) Channel T7

(b) Channel FP2

Figure 8: Visualization of DEAP spectrograms for different stimuli on T7 & T8 channels for an arbitrary subject.

$$\Delta y = \sum_{i,j} (x_{ij} - x_{ij}^0) w_{ij} \tag{10}$$

DeepLIFT then distributes proportionally to each input based on its contribution to this difference. Formally, each input's contribution is:

$$C_{\Delta x_{ij}} = \frac{(x_{ij} - x_{ij}^0) w_{ij}}{\sum_{p,q} (x_{pq} - x_{pq}^0) w_{pq} + \varepsilon} \Delta y \tag{11}$$

where $\varepsilon$ is a small stabilizing constant to avoid division by zero. For EEG signal analysis, we utilized a zero-level baseline input, meaning all elements in the baseline tensor are set to zero, enabling clear attribution of feature relevance relative to a neutral reference point.

## A.5 RESULTS EXPLANATION

### A.5.1 BASELINES

**TS-MoCo**: Hallgarten et al. (Hallgarten et al., 2023) proposed a self-supervised learning framework with momentum contrast and a transformer-based architecture consisting of a student and teacher context encoder, and a reconstruction head based on GRU.

**CLDTA**: Liao et al. (Liao et al., 2024) incorporated a position and source data embeddings with a diagonal masking strategy and an information separation technique inside a custom transformer-based architecture with self-unknown attention mechanism.

**SGMC**: Kan et al. (Kan et al., 2023) developed a genetics-inspired data augmentation method, named SGMC, which generates augmented groups by pairing, cross exchanging, and separating data samples, with further aggregation by a projector model to extract group-level features.

**GMSS**: Li et al. (Li et al., 2022) proposed a graph-based multi-task SSL model (GMSS) to learn representations by integrating multiple tasks, including frequency and spatial jigsaw puzzle tasks, and contrastive learning tasks.

**EEGformer**: Wan et al. (Wan et al., 2023) proposed a model with 1D-CNN and three transformer-based sequential encoders: regional, synchronous, and temporal transformers.

**MMResLSTM**: Ma et al. (Ma et al., 2019) proposed a multimodal residual LSTM (MMResLSTM) model, which contains the residual blocks and parallel LSTMs sharing weights among different multiple modalities.

**RGNN**: Zhong et al. (Zhong et al., 2020) proposed a regularized graph neural network (RGNN) with modeling biological topology among different brain regions via an adjacency matrix in a GNN and two regularizers: node-wise domain adversarial training and emotion-aware distribution learning.

**MSBAM**: Wu et al. (Wu et al., 2022) proposed a multi-scales bi-hemispheric asymmetric model (MSBAM) with baseline correction and 3D EEG transformation processing with spatial feature extractor block and bi-hemispheric asymmetric temporal feature extractor block.

### A.5.2 VISUALIZATION OF COMPRESSED EMBEDDINGS

The visualization of compressed embeddings both for SEED and DEAP datasets shown in Figure 9.

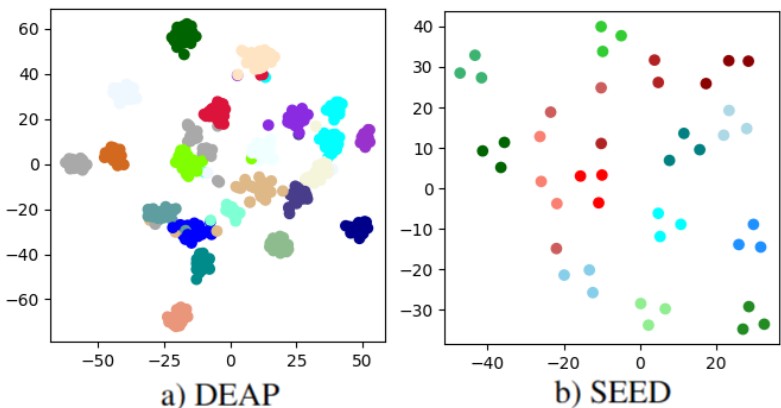

Figure 9: Visualization of embeddings compressed to 2 dimensions using PCA.

### A.5.3 FEATURE EXTRACTION TECHNIQUES COMPARISON

We compared results of our approach on the feature extraction techniques for SEED and DEAP datasets on subject-independent and subject-dependent splits. The accuracies in percents (%) are shown in Table 9. The feature extraction methods were evaluated separately for such techniques that process the whole video, and those making predictions by chunks of video, sliced one second at a time without overlapping. The second type includes raw data, raw data with baseline subtraction, and these views followed by spatial transformation (named 3D in the Tables).

For the SEED dataset, DE features showed the highest accuracy for the processing of the whole video, while raw data and baseline removal showed the highest results for the per-second processing. For the DEAP dataset, for the processing of the whole video , DE features performed best, with baseline removal making a small contribution to the subject-dependent score. Meanwhile, in the case of per-second processing, baseline correction was the most effective feature extraction technique showing 63% and 95% to subject-independent and subject-independent scores.

### A.5.4 MODEL TRAINING OPTIMIZATION COMPARISON USING ITERATIVE DEEPLIFT APPLICATION (IDA)

Here, we also report *downstream* training speed during each step of IDA (Figure 10), and demonstrate the full results of the optimization of the model training from the initial state to the last state (Table 10) that shows a high increase in model performance.

In addition to pruning EEG channels, we removed less relevant frequency bands. Complete results are reported in Table 11.

Table 9: Accuracy comparison on different feature extraction techniques

| Feature type | SEED | | DEAP | |
|---|---|---|---|---|
| | SI | SD | SI | SD |
| DE | **87.3** | **88.7** | **60.6** | 62.4 |
| Baseline + DE | 86.7 | 85.2 | **60.6** | **65** |
| PSD | 61.5 | 65 | 58.5 | 59 |
| Baseline + PSD | 57.8 | 53.7 | 56.3 | 62.4 |
| DASM | 64.4 | 58.6 | 54.2 | 52.4 |
| Baseline + DASM | 56.6 | 52.2 | 51.4 | 55.9 |
| RASM | 42.2 | 39.9 | 54.2 | 52.4 |
| Baseline + RASM | 40 | 39 | 57 | 52.8 |
| DCAU | 69.6 | 59.6 | 53.5 | 55.9 |
| Baseline + DCAU | 60.7 | 54.7 | 54.9 | 55.5 |
| Raw | **66.7** | 81.9 | 53.5 | 68.6 |
| Baseline | 63.9 | **91.2** | **63.1** | **95.3** |
| 3D | 56 | 66.4 | 55.3 | 68 |
| Baseline + 3D | 44.2 | 90.3 | 51.1 | 94.5 |

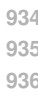


Figure 10: Model training speed on the downstream task for DEAP and SEED: Seconds per iteration vs. Number of EEG channels.

Table 10: Seconds per iteration of model training for pre-text and downstream tasks before and after EEG channel reduction.

| Dataset | Type | Pre-Text Init | Pre-Text Red | Downstream Init | Downstream Red |
|---|---|---|---|---|---|
| SEED DE | SD | 7.40 | 0.50 | 1.20 | 0.20 |
| | SI | 6.80 | 0.60 | 1.30 | 0.20 |
| SEED Baseline | SD | 58.10 | 4.80 | 19.20 | 2.10 |
| | SI | 55.70 | 4.60 | 18.50 | 2.20 |
| DEAP DE | SD | 1.20 | 0.30 | 0.30 | 0.10 |
| | SI | 1.20 | 0.30 | 0.30 | 0.10 |
| DEAP Baseline | SD | 39.80 | 9.00 | 9.00 | 2.70 |
| | SI | 42.60 | 8.90 | 9.80 | 2.80 |

Table 11: Channels, frequency bands, and accuracy (before/after) data reduction.

| Dataset | Type | Channels | Bands | Accuracy Init | Accuracy Red |
|---|---|---|---|---|---|
| SEED DE | SD | T7, T8 | Delta | 88 | 82 |
| | SI | T7, T8 | Delta | 87.3 | 84 |
| | SI | T7, T8 | Delta, Theta, Beta | 87.3 | 86 |
| SEED Baseline | SD | T7, T8 | — | 91.2 | 89 |
| | SI | FC1, T7, PZ, PO4 | — | 63.9 | 56 |
| DEAP DE | SD | T7, T8 | Theta | 62.4 | 59 |
| | SI | T7, T8 | Theta | 60.6 | 57 |
| DEAP Baseline | SD | T7, T8 | — | 95.3 | 89.5 |
| | SI | T7, T8 | — | 63.1 | 53.4 |

