# OpenReview forum: "CLIQ: Contrastive Learning with XAI-guided Interpretation and Model Quantization for EEG-based Emotion Recognition"
_ICLR.cc/2026/Conference — ICLR 2026 Conference Withdrawn Submission_

### Official Review · Reviewer_DjcT · 2025-10-30

**Soundness:** 2
**Presentation:** 2
**Contribution:** 2
**Rating:** 2
**Confidence:** 5

**Summary:**

This paper proposes a self-supervised EEG emotion recognition framework called CLIQ, designed to enhance classification accuracy and model efficiency through an improved contrastive learning strategy. Based on a convolutional encoder, the method introduces a modified negative sample construction and batching approach to better utilize limited labeled data. Experiments on SEED and DEAP datasets under both subject-dependent and subject-independent settings show that CLIQ achieves comparable or better performance than existing methods with fewer parameters and faster inference. The authors also employ feature visualization and post-training quantization to identify key EEG channels (e.g., T7, T8) and compress the model.
Overall, the paper aims to build an end-to-end and interpretable EEG emotion recognition system, but its methodological novelty and experimental depth remain limited, representing mainly an incremental adaptation of existing contrastive learning techniques.

**Strengths:**

This paper applies self-supervised learning to EEG-based emotion recognition, a topic of practical relevance. The authors emphasize model efficiency and deployability by exploring channel selection and post-training quantization, while introducing a soft–hard distinction for negative pairs in contrastive learning. Overall, the strengths lie in the practical significance of the research direction and the completeness of the engineering implementation.

**Weaknesses:**

1) The paper lacks clear structural and visual explanations of the proposed method. Key figures (e.g., Figure 2) are low-resolution raster images rather than vector graphics, making them blurry when enlarged. Moreover, the architecture diagram is overly simplified and does not adequately illustrate the relationships and data flow between core modules, which makes the overall framework difficult to follow.
2) The experiments are conducted only on two relatively small public datasets, SEED and DEAP, which is insufficient to support the paper’s general claims. The lack of validation on larger or more diverse datasets limits the generality and robustness of the conclusions. In addition, some parts of the experimental setup appear to reuse configurations or feature settings from prior work without ensuring strict consistency in preprocessing or evaluation, raising concerns about the fairness of the comparisons.
3) Based on the experimental description and partial code inspection, it seems that in the subject-independent (SI) setting, the same subject was fixed as the test set instead of performing random or cross-validated splits, which could introduce bias.
Furthermore, for the SEED dataset, if data splitting is done at the sample level rather than by trial, there is a high risk of data leakage that could inflate performance. Although the authors mention using 10-fold and Leave-One-Subject-Out validation, the exact splitting criteria and randomization strategy are not clearly stated, leaving the experimental reliability in question.
4) The proposed distinction between hard and soft negative pairs, with different weighting in the contrastive loss, is highly similar to existing approaches that perform sample re-weighting or hard-negative mining based on similarity or difficulty. The presented loss function essentially corresponds to a temperature-scaled weighted BCE formulation, without any new theoretical derivation or property analysis. Compared to the standard InfoNCE loss, the proposed modification appears to be an engineering tweak rather than a principled methodological innovation, and the reported performance gain is not substantial.

**Questions:**

1) Please clarify the train/test partitioning strategy used in the subject-independent (SI) experiments. Was the same subject fixed as the test set across all runs, or was a random or cross-validated split applied? In addition, for the SEED dataset, was the data split performed at the trial level or at the sample level? If it was the latter, there is a potential risk of data leakage between training and testing samples, which may affect the reliability of the reported results.
2) The paper only reports results on two datasets, SEED and DEAP, which limits the generalizability of the findings. Could the authors explain why additional public EEG emotion datasets such as SEED-IV, DREAMER, or AMIGOS were not included for further validation of the proposed model?
3) The proposed distinction between hard and soft negative pairs appears similar to existing approaches in contrastive learning that use sample re-weighting or hard-negative mining strategies. Could the authors elaborate on how their formulation differs fundamentally from prior work and provide empirical evidence that it offers measurable improvements over the standard InfoNCE loss?

---

### Official Review · Reviewer_KQv6 · 2025-10-31

**Soundness:** 3
**Presentation:** 3
**Contribution:** 3
**Rating:** 6
**Confidence:** 3

**Summary:**

The paper proposes CLIQ, a convolutional encoder trained with a novel contrastive framework for EEG-based emotion recognition. The work combines (1) multiple preprocessing pipelines (temporal, frequency, time frequency), (2) a modified contrastive loss that weights positive / soft-negative / hard-negative pairs, (3) a batch construction algorithm intended to guarantee representative pair composition, (4) feature analysis plus iterative DeepLIFT-based channel pruning, and (5) post-training symmetric quantization to shrink the model for edge deployment. Experiments on SEED and DEAP show competitive accuracy (e.g., SD/ SI numbers reported in Tables 1–2) together with large reductions in model size and training/inference time after pruning/quantization.

**Strengths:**

1. The paper addresses a relevant and practical problem: improving EEG-based emotion recognition while keeping models lightweight enough for edge devices. The combination of self-supervised learning, explainable pruning, and post-training quantization is clearly motivated by real deployment needs.
2. The experiments are comprehensive. The authors evaluate on two widely used datasets (SEED and DEAP).
3. Ablation studies are solid. The comparison between the proposed setup and a simpler baseline provides clear evidence that the new components contribute to performance gains.

**Weaknesses:**

1. α=β=0.4, γ=0.2, τ=0.5 lack sensitivity analysis. Add tests on parameter ranges (e.g., τ=0.1-1.0) and compare with equal weights.
2. Expand Eq. (1) with explicit indices, define $f_{ij}$​ precisely (e.g., cosine similarity scaled by $\tau$), and state how $\delta$ is assigned for positive/soft-/hard-negative pairs; brief pseudocode for pair/view formation would help reproducibility.

**Questions:**

1. How does CLIQ perform under cross-dataset transfer without fine-tuning?
2. How do the author ensure no subject/session leakage in contrastive pair construction and preprocessing?

---

### Official Review · Reviewer_XheM · 2025-11-01

**Soundness:** 2
**Presentation:** 2
**Contribution:** 2
**Rating:** 2
**Confidence:** 4

**Summary:**

This paper applies self-supervised learning (SSL) to process complex
EEG signals with low amount of labeled data for solving emotion recognition
task. Proposed approach is based on a convolutional encoder with a novel contrastive
loss and batching function. It has been evaluated on SEED and DEAP
datasets. The paper also compared different preprocessing techniques in temporal, frequency
and temporal-frequency domains. We achieved fairly high accuracy even
on small amount of labeled data

**Strengths:**

This paper proposes a self-supervised sentiment recognition method and achieves favorable results in both subject-dependent and subject-independent paradigms, demonstrating a certain degree of contribution.

**Weaknesses:**

1. The innovation of this paper is relatively weak, as it merely combines convolutional neural networks (CNNs) with contrastive loss in a straightforward manner.

2. The implementation of this paper lacks detailed ablation experiments, only including the baseline model and the final model. The performance of its baseline model is superior to that of most comparative models, while the improvement of the final model is limited.

3. The experiments on each dataset lack specific sentiment classification breakdowns.

**Questions:**

Please refer to the issues mentioned in the weaknesses section.

---

### Official Review · Reviewer_XYDD · 2025-11-05

**Soundness:** 2
**Presentation:** 3
**Contribution:** 2
**Rating:** 2
**Confidence:** 3

**Summary:**

This paper proposes a contrastive learning approach that integrates explainable artificial intelligence guided interpretation mechanisms with model quantization techniques for EEG-based emotion recognition. The study addresses the scarce annotation issue, high model complexity, and insufficient interpretability. The method was evaluated on two widely used datasets. Key EEG channels (T7 and T8) were identified through feature analysis and DeepLIFT, achieving competitive accuracy while reducing model size via quantization.

**Strengths:**

1. The ablation study confirms component efficacy. Cross-dataset transfer and embedding visualizations (t-SNE/PCA) test its generalizability.
2. T7/T8 identification matches literature on temporal lobe emotional processing, avoiding "black-box" limitations.
3. This paper achieves 4x memory reduction and faster inference, directly supporting edge-device deployment.

**Weaknesses:**

1. The batching algorithm sets parameter k=2 to "handle dataset variability", but provides no justification.
2. In cross-dataset transfer, the subject-independent accuracy when transferring from SEED to DEAP was only 63.4%, yet the authors did not conduct in-depth analysis to clarify the fundamental reasons behind this suboptimal result.
3. In Line 337, which mentions the biological relevance of the T7/T8 channel, lacks supporting citations, thereby weakening the claim of interpretability.
4. The CLIQ algorithm was not tested on signals containing simulated artifacts (such as muscle noise of varying intensities and eye movements) to assess whether its performance degrades.

**Questions:**

See weakness

---

### Note · Authors · 2025-11-21

I have read and agree with the venue's withdrawal policy on behalf of myself and my co-authors.